# Health facility preparedness for early detection of symptomatic cancer in Southern Africa: A multi-centre cross-sectional study

Tasleem Ras[1]*, Sarah Day[2,3], Bothwell Guzha[4], Valerie A. Sills[5], Suzanne E. Scott[5], Fiona M. Walter[5,6], Jennifer Moodley[2,3]

**1** Department of Family, Community and Emergency Care, Faculty of Health Sciences, University of Cape Town, South Africa, **2** Cancer Research Initiative, Faculty of Health Sciences, University of Cape Town, South Africa, **3** School of Public Health, Faculty of Health Sciences, University of Cape Town, South Africa, **4** Department of Obstetrics and Gynaecology, Faculty of Medicine & Health Sciences, University of Zimbabwe, **5** Wolfson Institute of Population Health, Faculty of Medicine and Dentistry, Queen Mary University of London, London, United Kingdom, **6** The Primary Care Unit, Department of Public Health and Primary Care, University of Cambridge, United Kingdom

☙ Senior authors
* tasleem.ras@uct.ac.za

## Abstract

Early detection and diagnosis of cancer in African contexts is challenging. Health services should identify people with symptoms suggestive of cancer, refer time-ously, and implement diagnostic processes efficiently. This requires the resources and infrastructure to provide these services. This study, the first in Southern Africa, reports on the availability of these services across referral pathways in four provinces of two Southern African countries, South Africa and Zimbabwe. As part of the African Awareness of Cancer and Early Diagnosis (AWACAN-ED) programme, we conducted a quantitative cross-sectional study, from February to September 2023 at primary care (PC), secondary and tertiary (ST) level public sector health facilities. PC's role is to identify and refer patients, while the definitive diagnosis mostly occurs at ST level. Data were collected on the availability of staff, infrastructure, diagnostic services, referral pathways and community engagement, and analysed descriptively. A total of thirty-four (N = 34) facilities were included. Twenty-two (n = 22) were PC facilities, serving a population of 1,068,177, with a nurse-to-1000-person ratio of 0.46, and doctor-to-1000-person ratio of 0.04. Twelve (n = 12) ST facilities were included, serving a population of 18,750,387, with a nurse-to-1000-person ratio of 0.42, and doctor-to-1000-person ratio of to 0.06. PC and ST facilities differed in availability of adequate communication infrastructure (40% vs 42%); patient transport systems (50% vs 100%); cervical smears/visual inspection (77% vs 100%) and colorectal clinical assessment (9% vs 58%). Mammography was not available at any Zimbabwean facilities. There was low availability of clinical protocols in primary care for cervical (30%) and breast (9%) cancer, and none for colorectal cancer. Community engagement activities focused on breast and cervical cancer. This study identifies areas for

**Data availability statement:** All relevant data are within the paper and its Supporting Information files.

**Funding:** This research was funded by the NIHR (NIHR133231) using UK international development funding from the UK Government to support global health research. TR, BG, SS, FW and JM were part-funded by the grant, while SD and VAS were fully funded by the grant for the duration of the project. No other funding source was used for this work. All authors contributed to the development of the research proposal including study tools. SD, JM and BG led the approval application processes. Once approved, TR had oversight of the data collection, while SD and BG co-ordinated the staff doing the data collection and ensured completeness of the dataset. TR led the data analysis, which was reviewed by the full team. TR wrote the first draft of this manuscript and the entire team reviewed.

**Competing interests:** The lead author (TR) was an associate editor for PLOS GPH from May 2024-March 2025. This did not alter our adherence to the PLOS Global Public Health policies on sharing data and materials. The authors have declared that no other competing interests exist.

improving early cancer diagnosis and provides a baseline for quality improvement interventions in facility infrastructure and referral pathways.

## Introduction and background

Cancer outcomes in Southern Africa are poor compared to those in high income countries (HIC) [1]. With an estimated 20 million new cases and 9.7 million deaths worldwide in 2022, the global burden of cancer is significant [2]. Observing trends of cancer over time suggests this global burden is expected to substantially worsen over the coming decades, with Africa expected to experience the fastest increase in cancer rates compared to other regions [3]. It is not known how prepared African health services are to shoulder this expected increased burden, or to spearhead interventions aimed at early diagnosis of common cancers.

Breast cancer is the leading cancer among women in Africa [3], and has an age-standardised mortality rate (ASMR) that is approximately 50% higher than global rates [4]. Cervical cancer, the second most common cancer among women in Africa [3], disproportionately affects low-middle income countries (LMIC) [5,6], and has an ASMR nearly three times the global average [7]. Colorectal cancer, which is the third most diagnosed cancer globally, is also a leading cause of cancer morbidity and mortality in Africa [8,9], with increasing incidence rates reported in South Africa (SA), Uganda and Zimbabwe [9]. The 5-year survival rate of colorectal cancer in Sub-Saharan Africa (SSA) is also low, with only 48% of individuals surviving for 5 years from the time of diagnosis [10]. One of the cardinal reasons for these high mortality rates is advanced-stage at the time of diagnosis [11–13].

A variety of factors contribute to diagnosis occurring at an advanced stage. Most patients with cancer first present with symptoms to primary care facilities, and patient factors such as low awareness, limited knowledge, or poor uptake of screening services often lead them to present at a late stage [12]. Also, health system factors such as limited availability of services or misdiagnosis contributes to the high mortality and low survival rates associated with breast, cervical and colorectal cancer in Africa [11–13]. The 'Model of Pathways to Treatment' offers a pragmatic approach to identifying patient, health system or disease factors that impact the timing of presentation to clinical services and hence clinical outcome [14,15]. In Southern Africa, health system factors such as high costs, poor care co-ordination and inadequate staffing have been found to be directly implicated in delayed diagnosis after patients present to health facilities, and in delayed referral to the appropriate treatment facilities [16]. South Africa has had national policies directing clinical care for breast and cervical cancer in place for a number of years [17,18]. These policies emphasise rapid referral to specialist care, and describe how health services should be organised to achieve this. It follows that properly organised, resourced and capacitated services could play a significant role in influencing cancer outcomes in Africa.

As part of the global strategy to enhance health outcomes across a wide range of diseases, the World Health Organisation (WHO) produced a framework for the organisation and delivery of health services [19]. This framework proposes that

certain key structures, inputs and processes are required if health services are to effectively deliver on the mandate of improved health outcomes. Key among these are policies, monitoring and evaluation, physical infrastructure, medicines and equipment, health information systems, effective care models with community linkages, and high-quality care at the first point of accessibility. In SA, this framework is mirrored in the Ideal Facility initiative, which provides guidelines for the organisation and functioning of all state-funded health facilities, and is monitored by the Office of Health Standards and Compliance (OHSC) [20]. Zimbabwe has a similar system in place, monitored by the Health Professions Authority (HPA) via their Inspectorate Division, which publishes and enforces minimum requirements for all health facilities [21]. The documents published by these three agencies (WHO, OHSC, HPA) are comprehensive in their scope and provide a basis for planning, implementing and evaluating cancer-related health services. However, an assessment of health service readiness for early detection and diagnosis of cancer has yet to be undertaken. Both countries have mixed public/private health services, with the first point of contact being primary level clinics, who refer to district hospitals, and then to specialised hospitals (secondary, tertiary or quaternary) [22,23].

This study aimed to evaluate the preparedness of health facilities to facilitate early detection and diagnosis of three cancers (breast, cervical and colorectal) commonly found in two Southern African countries, SA and Zimbabwe. This is part of a NIHR-funded African Awareness of Cancer and Early Diagnosis (AWACAN-ED) research programme [24].

## Methods

We conducted a multi-centre, cross-sectional, descriptive study spanning primary, secondary and tertiary public health facilities in two regions of SA and Zimbabwe.

### Study setting

In SA, two provinces were included: the Eastern Cape (SA-EC) and Western Cape (SA-WC). The SA-EC is more rural and less resourced (industrially and infrastructurally) than the SA-WC. Referral pathways that incorporated primary, secondary and tertiary level services constituted a distinct unit of analysis within each region, with primary care (PC) facilities situated in rural and urban communities. Typically, patients with possible cancer symptoms self-present to PC clinics and are assessed by a nurse, although some clinics might have access to a doctor. Patients assessed as having possible cancer symptoms are then referred to specialist secondary level regional/district hospitals for investigation and definitive diagnosis. In SA, urban clinics are typically less than 30 minutes by road to the referral hospital, while the travelling time for rural patients varies from 1-3 hours. Depending on resources (mammography, colonoscopy, imaging facilities, general surgeons, specialist surgeons, etc.) patients might be diagnosed and treated at this level or referred to a tertiary level facility for further investigation and treatment as secondary level hospitals do not have radiotherapy or chemotherapy services. In some cases, where there are no intervening secondary-level hospitals, patients may be referred directly from PC to outpatient clinics in tertiary facilities.

In Zimbabwe, Harare (Zim-H) and Bulawayo (Zim-B) provinces were included. Zimbabweans residing in the Zim-H and four surrounding provinces are served by three tertiary hospitals, and those residing in the Zim-B region are served by two tertiary hospitals. In these two metropolitan provinces there are no secondary health care facilities; therefore, patients with possible cancer symptoms who self-present to nurse-led PC clinics, are referred directly to the tertiary hospitals. Travelling times from urban PC to referral hospitals are similar to SA, less than 30 minutes, while rural-situated patients could travel for up to six hours by road to get to the referral hospital. Once the cancer diagnosis has been confirmed, patients receive specialised diagnostic investigations and treatment at their geographically appropriate tertiary hospitals.

### Sampling

The sampling frame for each of the four geographic regions consisted of all primary, secondary and tertiary health facilities within a specified clinical referral pathway. PC health facilities were sampled purposively, ensuring a mix of rural and

urban facilities. All secondary and tertiary facilities within the designated referral pathway were included. For this paper, we used the working definition of Statistics South Africa (StatsSA) of what constitutes a rural area: sparsely populated, with agriculture or mining being the dominant means of livelihood [25]. In PC the participants were facility managers, and in secondary/tertiary hospitals they were operational managers or lead clinicians.

## Data Collection

We developed a comprehensive Health Facility Assessment (HFA) tool based on local and international guidelines plus input from our collaborating team including cancer researchers, public health specialists, PC specialists, and cancer-specific specialists from Zimbabwe, SA and the UK (S1 Appendix) [16–20]. The HFA tool covered ten domains that describe essential elements of a health service, aligned broadly with the WHO's 'building blocks' (see Table 1). Each of these domains had items that directly describes clinical services. Because the focus of this study was on cancer diagnosis only (and not treatment), it was deemed appropriate to pool the specialised diagnostic services provided by secondary and tertiary hospital into one category (ST - secondary/tertiary). Additionally, mammography, colposcopy, colonoscopy and core biopsy were excluded for PC as these were deemed not appropriate in these contexts. From this rationalisation process, two distinct tools were produced, one for PC and one for ST. These two tools were then converted to electronic format in REdCap, a secure online application used for building and managing databases [26]. This digital format was then used to train the data collectors, who subsequently piloted the data collection process on eight facilities (four in each country) that were not part of the study sample. This led to modifications in the structure of some items, refinement of the options offered to respondents, and extra training for data collectors.

Data were collected, between 01-02-2023 and 01-10-2023, by in-person structured interviews led by a field worker or, if specifically requested by participants, the data collection tool was sent electronically or physically to be completed independently and later entered onto REDCap. All interviews took place in English and lasted approximately sixty minutes. If answers were not immediately available or all questions were completely addressed, the data collectors would liaise with the respondent on a separate occasion to ensure each facility dataset was complete.

Data was exported into Excel, cleaned, and again checked for completeness per facility. Facilities were categorised according to their geographic location and clinical level (PC vs ST).

The data were analysed descriptively and disaggregated according to the geographic location and clinical level at which they were collected. Results are described per facility, and where appropriate, as a proportion of the total number of facilities.

## Ethical considerations

The protocol for this study was reviewed and approved by institutional review boards of the University of Zimbabwe (JREC ref.: 363/21) and the University of Cape Town (UCT HREC ref.: 664/2021). Subsequently, written approval was obtained from the Medical Research Council of Zimbabwe (MRCZ/A/2831) and the relevant SA provincial health authorities to implement the study in the identified facilities. Additionally, written informed consent was obtained from all participants prior to commencement of the data collection. Data was de-identified during the cleaning phase, rendering the facilities and participants anonymous.

## Results

A total of thirty-four facilities were included across SA and Zimbabwe. Of these, twenty-two were PC facilities (SA-WC n = 4, SA-EC n = 3, Zim-H n = 9, Zim-B n = 6). Five were defined as rural (SA-WC n = 1, SA-EC n = 3, Zim-B n = 1). A total of twelve secondary/tertiary facilities were included, spread across the four provinces (SA-WC n = 2, SA-EC n = 2, Zim-H n = 5, Zim-B n = 3). Two of these self-defined as rural (SA-EC n = 1, Zim-B n = 1).

The catchment population of the selected geographic regions are shown in Table 2 below.

**Table 1. Data collected using *Health Facility Assessment* (*HFA*) tool.**

| Domains | Items |
|---|---|
| 1. Facility information | 1. Facility identification<br>2. Geographic co-ordinates<br>3. Catchment population |
| 2. Staffing | 1. 2. Number of staff involved in cancer-related services<br>Cancer-specific training |
| 3. General facility infrastructure | 1. Communication access: fixed landline; mobile telephone; email; internet; computer<br>2. Power supply: electricity supply; emergency back-up generator<br>3. Water supply: clean piped water<br>4. Ambulance: emergency; elective<br>5. General equipment: refrigerator; autoclave<br>6. Personal Protective Equipment: soap; gloves; sanitizer; goggles; aprons; towels; masks<br>7. Infection Prevention and Control: written policy; implementation monitoring<br>8. Waste disposal: bins; sharps; biological waste |
| 4. Specific infrastructure and equipment | 1. Ultrasound machine<br>2. Biopsy- fine needle<br>3. Vaginal speculum<br>4. Lithotomy bed<br>5. Examination lamp<br>6. For secondary/tertiary:<br>7. Colposcopy machine<br>8. Colonoscope machine<br>9. Mammography machine<br>10. Ultrasound machine<br>11. Biopsy - core<br>12. Biopsy – fine needle<br>13. Biopsy – Punch |
| 5. Cancer diagnostic services | 1. Clinical assessment<br>2. Cervical smears/liquid based cytology<br>3. Visual inspection of cervix<br>For secondary/tertiary (additional):<br>4. Mammography<br>5. Colposcopy<br>6. Colonoscopy |
| 6. Medical record system | Electronic or paper |
| 7. Referral systems and protocols | 1. Clinical protocols for diagnosis<br>2. Clinical protocols for referral/receipt<br>3. Referral register |
| 8. Transport to diagnostic services | Availability, type |
| 9. Feedback systems | Presence and form of feedback, e.g., telephone, written, electronic |
| 10. Community outreach | 1. Education and awareness activities<br>2. Community-based screening |

## Facility profile by staffing, catchment population and cancer-related training

**Primary care (PC).** The twenty-two PC facilities serve an estimated 1,068,177 people. The full details of the catchment population, staffing, and training of each facility are available in S2 Appendix. Four hundred and forty-eight nurses and 42 doctors were employed at these facilities, giving an overall nurse-to-1000 persons ratio of 0.42 (range: 0.1-1.18). The

**Table 2. Catchment population of geographic regions.**

| Geographic area | Total population | Age groups | Population | % female |
|---|---|---|---|---|
| *South Africa* | | | | |
| Western Cape | 7 433 020 | 15-19 | 532 982 | 51.4 |
| | | 20-29 | 1 344 059 | 50.0 |
| | | 30-39 | 1 360 304 | 50.1 |
| | | 40-49 | 990 685 | 51.1 |
| | | 50-59 | 719 070 | 54.4 |
| | | 60+ | 818 399 | 70.4 |
| Eastern Cape | 7 230 204 | 15-19 | 665 744 | 48.5 |
| | | 20-29 | 1 149 387 | 50.1 |
| | | 30-39 | 1 016 788 | 52.1 |
| | | 40-49 | 761 616 | 54.2 |
| | | 50-59 | 629 705 | 58.9 |
| | | 60+ | 766 646 | 71.3 |
| *Zimbabwe* | | | | |
| Harare | 2 487 209 | 15-19 | 481 886 | 51.8 |
| | | 20-29 | 923 630 | 53.6 |
| | | 30-39 | 660 212 | 49.7 |
| | | 40-49 | 328 002 | 47.8 |
| | | 50-59 | 208 113 | 56.8 |
| | | 60+ | 237 415 | 54.3 |
| Bulawayo | 663 940 | 15-19 | 75 253 | 53.4 |
| | | 20-29 | 137 317 | 55.8 |
| | | 30-39 | 89 655 | 52.5 |
| | | 40-49 | 54 099 | 52.9 |
| | | 50-59 | 38 937 | 53.3 |
| | | 60+ | 32 759 | 56.0 |

overall doctor-per-1000 persons ratio was 0.04 (range: 0.0-0.2). Notably, the PC facilities in SA-EC and Zim-H reported no doctors on their staff establishments. Zim-H had the lowest regional nurse-per-1000 persons ratio of 0.2 (range: 0.1-0.5). The highest regional nurse-per-1000 persons ratio of 0.9 (0.7-1.7) was reported in Zim-B. Most facilities (16/22) reported that cancer-related in-service training occurred, with all providing breast and cervical examination or smear training, while none reported training in colorectal cancer assessment.

**Secondary/Tertiary (ST).** The twelve ST facilities we assessed provided services to a total estimated population of 18,750,387 people with a total in-patient bed capacity of 6,896. Catchment population, staffing and training details are listed in S2 Appendix. The overall nurse-per-1000 persons ratio was 0.3 (range: 0.05-6.6), while the overall doctor per 1000 people ratio was 0.06 (range: 0.01-0.23). Zim-H had the highest regional nurse-per-1000 persons ratio of 0.7 (0.6-6.6), and SA-WC had the highest doctor-per-1000 persons ratio of 0.14 (0.07-0.16). SA-EC had the lowest nurse-per-1000 persons ratio (0.1: 0.05-0.3)) and the lowest doctor-per-1000 persons ratio of 0.02 (0.01-0.04).

## Availability of general infrastructure

**Primary care (PC).** The twenty-two items that describe general infrastructure are shown in Fig 1. In general, SA PC facilities had better health infrastructure compared to Zimbabwean PC facilities.

**Fig 1. General Infrastructure in primary care facilities.**

| Facility | Comms | Power | Water | Amb | Equip. | PPE | Waste | Total |
|---|---|---|---|---|---|---|---|---|
| ...ssible score | 5 | 2 | 1 | 2 | 2 | 7 | 3 | 22 |
| **SA - Western Cape** | | | | | | | | |
| SAWCPC1 | 5 | 2 | 1 | 2 | 2 | 7 | 3 | 22 |
| SAWCPC2 | 5 | 2 | 1 | 2 | 2 | 7 | 3 | 22 |
| SAWCPC3 | 4 | 2 | 1 | 2 | 2 | 7 | 3 | 21 |
| SAWCPC4 | 3 | 2 | 1 | 2 | 2 | 7 | 3 | 20 |
| **SA - Eastern Cape** | | | | | | | | |
| SAECPC1 | 0 | 1 | 1 | 0 | 2 | 7 | 3 | 14 |
| SAECPC2 | 3 | 2 | 1 | 0 | 1 | 6 | 3 | 16 |
| SAECPC3 | 5 | 2 | 1 | 1 | 1 | 6 | 3 | 19 |
| **Zimbabwe - Harare** | | | | | | | | |
| ZHPC1 | 2 | 1 | 1 | 0 | 1 | 4 | 2 | 11 |
| ZHPC2 | 2 | 1 | 1 | 2 | 2 | 4 | 3 | 15 |
| ZHPC3 | 0 | 2 | 1 | 2 | 2 | 4 | 3 | 14 |
| ZHPC4 | 1 | 2 | 1 | 0 | 2 | 5 | 3 | 14 |
| ZHPC5 | 1 | 2 | 1 | 2 | 1 | 5 | 3 | 15 |
| ZHPC6 | 1 | 2 | 1 | 0 | 1 | 5 | 3 | 13 |
| ZHPC7 | 0 | 1 | 1 | 0 | 1 | 6 | 3 | 12 |
| ZHPC8 | 1 | 1 | 1 | 2 | 1 | 4 | 3 | 13 |
| ZHPC9 | 3 | 2 | 1 | 1 | 2 | 5 | 3 | 17 |
| **Zimbabwe - Bulawayo** | | | | | | | | |
| ZBPC1 | 5 | 2 | 1 | 2 | 2 | 7 | 3 | 22 |
| ZBPC2 | 2 | 2 | 1 | 0 | 2 | 5 | 3 | 15 |
| ZBPC3 | 1 | 2 | 1 | 1 | 2 | 6 | 3 | 16 |
| ZBPC4 | 5 | 2 | 1 | 0 | 2 | 7 | 3 | 20 |
| ZBPC5 | 1 | 2 | 1 | 0 | 1 | 6 | 3 | 14 |
| ZBPC6 | 1 | 2 | 1 | 0 | 2 | 5 | 3 | 14 |

*SAWCPC*: South Africa, Western Cape, Primary Care; *SAECPC*: South Africa, Eastern Cape, Primary Care; *ZHPC*: Zimbabwe, Harare, Primary Care; *ZBPC*: Zimbabwe, Bulawayo, Primary Care

**Domain legend:** *Comms:* communication infrastructure (landline, mobile, email, internet, computer); *Power:* Access to electricity and back-up generator; *Water:* access to clean piped water; *Amb:* access to ambulance services for emergency and elective patient transport; *Equip:* Equipment (refrigerator and autoclave) needed for basic services; *PPE:* Personal Protective Equipment (soap, gloves, sanitiser, goggles, aprons, towels, masks); *IPC:* Infection Prevention and Control (policy and implementation monitoring); *Waste* (Bins, sharps, biological waste disposal)

Facilities performed variably in the communication infrastructure domain. While SA-WC facilities generally had access to most forms of communication, facilities in SA-EC and both Zimbabwean regions were limited in their access to fixed telephone lines, email and internet.

All facilities fared well in the power and water supply domains. Most facilities (17/22) had access to emergency backup electrical supply in case of any power outages.

Nine facilities had access to emergency and elective ambulance services, with eight of these located in an urban setting. Three facilities had access to emergency services only, while the balance (n = 10) had no access to ambulance services. Notably, the largely rural regions of SA-EC and Zim-Bulawayo had constrained access (Fig 1).

When measuring access to personal protective equipment, twelve facilities had six or more items available, with the lowest score being measured in four facilities, all within the Zim-H province. The items often found to be unavailable were disposable towels (15/22), followed by goggles (8/22) and disinfectant (6/22).

All facilities had access to medical waste disposal.

**Secondary and tertiary (ST) care infrastructure.** Generally, ST hospitals had better general infrastructure than the PC facilities. The general infrastructural components (n = 24) for secondary/tertiary facilities are presented in Fig 2.

Communication infrastructure access varied amongst the twelve ST facilities, with most SA facilities scoring well in this domain. Most departments had access to a fixed telephone line (9/12), while computers (5/12), email access (6/12) and internet access (7/12) were not as readily available. Notably, three Zimbabwean facilities reported no availability of communication infrastructure.

Elective ambulance services were available to most facilities (9/12), while emergency ambulances were more commonly available (11/12). Cleaned piped water, personal protective equipment, medical waste services, and infection prevention and control (IPC) were available at all facilities.

## Specific equipment and services for cancer assessment and diagnosis

**Primary care (PC).** The availability of equipment and services per facility is shown in Table 3 below. A full report of all measured items is available in S3 Appendix.

For those services specific to breast cancer, clinical breast examination was the key indicator of a diagnostic service on offer. This was provided at most facilities (17/22). Additionally, three facilities indicated that they had ultrasound services available on site, while two reported that they had equipment to perform biopsies of suspicious lesions.

While some form of assessment for cervical cancer was reported in most facilities (18/22), the type of assessment varied between countries. In SA facilities, cervical screening using smear/liquid-based cytology was the most common cervical assessment method reported (6/7) with two also reporting visual inspection with acetic acid (VIA), while in Zimbabwean regions the assessment of choice was either a clinical examination (11/22) or VIA (11/22). However, this was not

| Facility | Comms | Power | Water | Ambu | Equipm | PPE | IPC | Waste | Total |
|---|---|---|---|---|---|---|---|---|---|
| Possible score | 5 | 2 | 1 | 2 | 2 | 7 | 2 | 3 | 24 |
| **SA - Western Cape** | | | | | | | | | |
| SAWCH1 | 5 | 2 | 1 | 2 | 2 | 7 | 2 | 3 | 24 |
| SAWCH2 | 5 | 2 | 1 | 2 | 2 | 7 | 2 | 3 | 24 |
| **SA - Eastern Cape** | | | | | | | | | |
| SAECH1 | 4 | 2 | 1 | 2 | 2 | 7 | 2 | 3 | 23 |
| SAECH2 | 2 | 2 | 1 | 2 | 2 | 6 | 2 | 2 | 19 |
| **Zimbabwe - Harare** | | | | | | | | | |
| ZHH1 | 3 | 2 | 1 | 2 | 2 | 7 | 2 | 3 | 22 |
| ZHH2 | 0 | 2 | 1 | 1 | 2 | 6 | 2 | 3 | 17 |
| ZHH3 | 4 | 2 | 1 | 2 | 2 | 7 | 2 | 3 | 23 |
| ZHH4 | 2 | 2 | 1 | 2 | 2 | 7 | 2 | 3 | 21 |
| ZHH5 | 0 | 2 | 1 | 1 | 2 | 7 | 2 | 3 | 18 |
| **Zimbabwe - Bulawayo** | | | | | | | | | |
| ZBH1 | 1 | 2 | 1 | 2 | 2 | 7 | 2 | 3 | 20 |
| ZBH2 | 2 | 2 | 1 | 1 | 2 | 6 | 2 | 3 | 19 |
| ZBH3 | 0 | 2 | 1 | 1 | 2 | 4 | 2 | 3 | 15 |

**SAWCH**: South Africa, Western Cape, Hospital; **SAECH**: South Africa, Eastern Cape, Hospital; **ZHPH**: Zimbabwe, Harare, Hospital;; **ZBPH**: Zimbabwe, Bulawayo, Hospital

**Domain legend: Comms:** communication infrastructure (landline, mobile, email, internet, computer); **Power:** Access to electricity and back-up generator; **Water:** access to clean piped water; **Amb:** access to ambulance services for emergency and elective patient transport; **Equip:** Equipment (refrigerator and autoclave) needed for basic services; **PPE:** Personal Protective Equipment (soap, gloves, sanitiser, goggles, aprons, towels, masks); **IPC:** Infection Prevention and Control (policy and implementation monitoring); **Waste** (Bins, sharps, biological waste disposal)

**Fig 2. General Infrastructure at Secondary/Tertiary Hospitals.**

Table 3. Cancer assessment and diagnostic services in primary care facilities.

| Facility | Breast Clinical assessment | Cervix Clinical assessment | Pap smear | VIA | Colo-rectal Clinical assessment |
|---|---|---|---|---|---|
| SA – western cape | | | | | |
| SAWCPC1 | ✓ | | ✓ | | ✓ |
| SAWCPC2 | ✓ | | ✓ | | |
| SAWCPC3 | ✓ | | ✓ | ✓ | |
| SAWCPC4 | ✓ | ✓ | ✓ | | ✓ |
| SA – eastern cape | | | | | |
| SAECPC1 | ✓ | | ✓ | ✓ | |
| SAECPC2 | | | | | |
| SAECPC3 | ✓ | | ✓ | | |
| zim - harare | | | | | |
| ZHPC1 | ✓ | | | | |
| ZHPC2 | ✓ | ✓ | | | |
| ZHPC3 | ✓ | ✓ | | ✓ | |
| ZHPC4 | | ✓ | | ✓ | |
| ZHPC5 | ✓ | ✓ | | ✓ | |
| ZHPC6 | | | | ✓ | |
| ZHPC7 | | | | | |
| ZHPC8 | ✓ | ✓ | ✓ | ✓ | |
| ZHPC9 | | ✓ | | ✓ | |
| zim - bulawayo | | | | | |
| ZBPC1 | ✓ | ✓ | | ✓ | |
| ZBPC2 | ✓ | | | | |
| ZBPC3 | ✓ | ✓ | | ✓ | |
| ZBPC4 | ✓ | ✓ | | ✓ | |
| ZBPC5 | ✓ | ✓ | | ✓ | |
| ZBPC6 | ✓ | ✓ | | ✓ | |

✓ = service provided

SAWCPC: South Africa, Western Cape, Primary Care; SAECPC: South Africa, Eastern Cape, Primary Care; ZHPC: Zimbabwe, Harare, Primary Care; ZBPC: Zimbabwe, Bulawayo, Primary Care.

always supported by the availability of the necessary equipment, with a vaginal speculum only being available in half the facilities (11/22), and an examination lamp even less (8/22).

Only two facilities, both in SA-WC, offered an assessment (including clinical examinations) for patients with suspected colorectal cancer.

Six facilities (all SA) reported that they were able to take specimens and send them to a laboratory for analysis. Seven facilities (SA n = 3; Zim n = 4) report having access to radiology services, with two having these services available on site.

**Secondary and tertiary (ST).** Information on the readiness of the twelve secondary/tertiary facilities to provide diagnostic services are presented below, in Table 4 and in S3 Appendix.

Breast cancer services were variably available across the regions, with mammography being notably absent across all Zimbabwean facilities. Only one hospital in Zim-B was able to perform tissue biopsies on suspicious lesions, this despite the equipment for core (9/12) and punch (10/12) biopsies being available in most facilities. Most of these facilities (11/12) had access to onsite ultrasound scanning services.

**Table 4. Cancer assessment and diagnostic services in Secondary/Tertiary facilities.**

| Facility | Breast | | | Cervix | | | | Colo-rectal | tumor board |
|---|---|---|---|---|---|---|---|---|---|
| | Clinical assess | Mammo-graphy | Biopsy | Clinical assess | Pap smear | VIA | Colposc-opy | Colonos-copy | |
| SA – WESTERN CAPE | | | | | | | | | |
| sawch1 | ✓ | ✓ | ✓ | ✓ | ✓ | | ✓ | ✓ | ✓ |
| sawch2 | ✓ | | ✓ | ✓ | ✓ | ✓ | ✓ | ✓ | |
| SA – EASTERN CAPE | | | | | | | | | |
| saech1 | ✓ | ✓ | ✓ | ✓ | ✓ | ✓ | ✓ | ✓ | |
| saech2 | ✓ | ✓ | ✓ | ✓ | ✓ | ✓ | ✓ | ✓ | |
| ZIM – HARARE | | | | | | | | | |
| zhh1 | ✓ | | ✓ | ✓ | ✓ | ✓ | ✓ | ✓ | |
| zhh2 | ✓ | | ✓ | ✓ | ✓ | ✓ | ✓ | | |
| zhh3 | ✓ | | ✓ | ✓ | ✓ | ✓ | ✓ | ✓ | ✓ |
| zhh4 | ✓ | | ✓ | ✓ | ✓ | ✓ | ✓ | | |
| zhh5 | ✓ | | ✓ | ✓ | | ✓ | | | |
| ZIM – BULAWAYO | | | | | | | | | |
| zbh1 | ✓ | | | ✓ | | ✓ | | | |
| zbh2 | ✓ | | ✓ | ✓ | | ✓ | | ✓ | |
| zbh3 | ✓ | | | ✓ | | ✓ | | | |

✓ = service provided.

SAWCH: South Africa, Western Cape, Hospital; SAECH: South Africa, Eastern Cape, Hospital; ZHPH: Zimbabwe, Harare, Hospital; ZBPH: Zimbabwe, Bulawayo, Hospital.

Cervical cancer diagnostic services, specifically colposcopy services, were widely available in the two SA regions and Zim-H (8/12) but absent in Zim-B, despite one facility in Zim-B reporting the availability of the necessary equipment.

Colorectal cancer services, indicated by the presence of colonoscopy services, were available in all four regions, though not in all facilities (7/12).

Pathology services were available to all these facilities, with half (6/12) having pathology services on site. Similarly, all facilities were able to provide radiology services, while most (11/12) provide ultrasound services. Only two facilities reported a functioning tumour board.

## Clinical guidance for diagnosis and referral

Only two of the twenty-two PC facilities had diagnostic protocols in place to assist clinicians in their clinical decision-making for breast cancer, both in SA-WC. A small number of PC facilities (4/22: one in SA-EC, two in Zim-H and one in Zim-B) had a referral protocol in place for suspected breast cancer, with the same number of facilities (n = 4: three in SA-WC and one in SA-EC) having a register available that recorded referral of such patients to higher levels of care.

For cervical cancer, some PC facilities (8/22: two in SA-WC, one in SA-EC, three in Zim-Harare, two in Zim-Bulawayo) reported having a diagnostic protocol in place. Half of the PC facilities (11/22) reported that they had referral protocols for suspected cervical cancer patients, and the same number (11/22) held a referral register for these patients.

One facility (SA-WC) reported having a colorectal cancer diagnostic and referral protocol. None of the facilities kept a referral register for patients with suspected colorectal cancer.

Nineteen facilities use paper-based referral processes, while the remaining three, all in SA-WC, used an electronic referral platform. This was also true for the use of electronic medical records (EMR), with all facilities reporting

paper-based records, except for the SA-WC facilities who indicated partial use of EMR for laboratory results, radiology reports and discharge summaries, while still relying on paper-based means to record daily clinical care.

Of the ST hospitals surveyed, all (12/12) received referred patients with suspected breast or cervical cancer, and most (11/12) accepted referred patients with suspected colorectal cancer. Not all facilities had formal written referral protocols in place: eight had a cervical cancer referral protocol, seven had a breast cancer referral protocol, and five had one for colorectal cancer. Diagnostic protocols fared similarly: nine facilities had protocols in place for cervical cancer, eight had breast cancer protocols in place, while five had protocols for colorectal cancer. Five hospitals kept a breast cancer register, eight had one for cervix cancer, while four had a colorectal cancer register.

### Cancer-related community engagement

Among the twenty-two PC facilities, cancer-related community engagements scored variably. Community engagement was defined as any activity of the facility staff that engaged with community-based organisations or directly with community members outside the physical structure of the health facility. Breast cancer-focussed community engagement was low, with one facility offering a community-based mobile screening service. Six facilities reported being involved in community-based breast cancer awareness programmes, with three of these, all based in SA, reporting ongoing active collaboration with community-based organisations.

In relation to cervical cancer, six PC facilities reported offering mobile screening services. Eight facilities were involved in community-based awareness programmes. Fourteen reported ongoing and active collaboration with a community-based organisation, with eleven of these located in Zimbabwe.

There were no mobile screening or community collaborations among PC facilities in relation to colorectal cancer, with one facility in SA reporting a community-based awareness activity.

The data relating to community engagement in the ST cohort showed similar gaps as the PC cohort.

In terms of breast cancer, four ST facilities in Zimbabwe reported providing an outreach diagnostic service for breast cancer. Eight facilities reported their involvement in breast cancer awareness programmes at community level (SA n = 3; Zim n = 5), while three reported breast cancer-related collaborations with community-based organisations. The situation for cervical cancer was similar: four facilities reported providing mobile diagnostic services; nine (SA n = 4; Zim n = 5) provided community-based awareness programmes; and five (SA n = 1; Zim n = 4) reported collaborating with community organisations. There was less colorectal cancer-related community-focussed activity, with two facilities providing outreach diagnostic services, four facilities providing community-based education, and one reported collaboration with a community organisation.

## Discussion

This is the first study to describe core elements in cancer-related assessment and diagnostic health services within four referral pathways in Southern Africa. While reporting on some positive aspects, scarcity of some essential elements is also described, especially in the PC facilities. This represents a baseline for future quality improvement interventions in these areas. Future research should explore technology-linked innovations to overcome the infrastructural challenges and continue generating data that supports advocacy for system strengthening.

The infrastructural and services gaps were not unexpected in this context. Hysia and colleagues have similarly described limitations in general and specific infrastructure in surgical and emergency services across five African countries [27]. Gaps in health infrastructure has been shown to have a causal relationship with health outcomes, as shown by Osakede who employed a Systems Gaussian Mixture Model (SGMM) to demonstrate this across multiple African settings [28]. WHO also acknowledges this in the recently published health systems evaluation framework, that highlights the importance of general and specific infrastructure in delivering comprehensive healthcare [16]. This WHO framework

could be used to develop consensus on essential elements of early cancer diagnostic interventions within low-resourced settings.

Communication infrastructure was identified as a key element of health services with significant gaps. The WHO Primary Health Care monitoring framework explicitly states that digital and communication technologies for health and health information systems are essential inputs for delivering on the desired health outcomes [16]. The gaps in communication infrastructure identified in our study marginalises remotely situated PC facilities within these health systems even further, limiting their capacity to connect with higher levels of care, which is essential to providing comprehensive health services. The use of mobile technologies not dependent on fixed line infrastructure may be a useful short-term intervention, though early evidence from various African contexts indicates significant challenges that still need to be addressed in this arena [29–31]. The challenges identified by these studies included limited mobile coverage, low technological skills amongst healthcare workers, and inconsistent power supply to remote areas. The persistent use of paper-based records as the main means of communication within and between facilities offers another area for possible technological intervention and efficiency gains, with subsequent system strengthening implications. Designing context-specific mobile applications, training healthcare workers in the integration of communication devices into their clinical duties and ensuring a steady power supply were some pragmatic and feasible solutions proposed to bridge the infrastructural gap.

The gaps in colorectal cancer diagnostic services were notable in that only 9% (2/22) of PC facilities indicated that a clinical examination was offered as an assessment, while 58% (7/12) of ST facilities offered colonoscopy services. In one Zimbabwean province, no colonoscopy services were available at all. This represents an area that requires significant attention, capacity-building and resource allocation, given the high prevalence [8,9] and mortality rates associated with advanced colorectal cancer at diagnosis in Africa [10–13]. The use of targeted faecal immunohistochemical tests (FIT) as a non-invasive accurate test, may assist remote PC facilities in identifying symptomatic patients needing rapid escalation of care [32–34]. In the context of low-resourced PC, introducing FIT would need careful exploration from the perspectives of the health system, clinical application and economic feasibility. This innovation should complement a focus on training healthcare workers on clinical examination techniques of the anus, rectum and colon.

We identified several challenges relating to the referral pathways for cancer patients. This included the absence of diagnostic protocols, referral criteria, and the necessary infrastructure (communication and transport) needed to co-ordinate care between PC and ST facilities. Providing high-quality, comprehensive primary care is a difficult task, and the absence of clear clinical protocols potentially compounds this difficulty, given the high workload and complex demands of PC in Southern Africa [35]. When local teams collaborate on co-creating referral pathways with consensus on referral criteria, clearly defined actions that follow decision-making, and robust communication processes, the chances of improved clinical care and patient outcomes are improved [36–38]. Encouraging clinical teams within specific referral pathways to collaborate around the urgency of early cancer diagnosis, negotiating for their autonomy in defining localised best practice, and facilitating enhanced skills in clinical quality improvement methods could offer promising results for health system strengthening across levels of care.

The seemingly low staffing in relation to population across some of these regions deserves more attention. This is not a unique finding, being foreshadowed by previous studies that have shown similar staffing in SA health systems for the past fifteen years [39,40]. The WHO recommends moving away from simple staff to population ratios, as described here, towards a more nuanced and contextually relevant system of calculating staffing requirements using the Workload Indicators of Staffing Needs (WISN) process [41]. The WISN, developed by the WHO and revised in 2010, is a multi-step, comprehensive method that identifies the needs of the population, assesses care complexity, time and skills required to provide comprehensive services, and rationalises allocation of human resources based on these factors. An example of applying this model proposed a staffing norm of 3 doctors per 5000 people (0.6 per 1000 persons) for primary care

centres in Oman [42], well above the findings we presented in this paper. While beyond the scope of this study, the WISN process may be an important next step for health system researchers, planners and leaders in these regions who are looking to invigorate PC health systems and cancer care.

## Policy implications and recommendations

We make the following recommendations that could enhance the delivery of cancer services in these regions.

1. Health authorities should use the WHO PHC monitoring framework to improve general and cancer-specific infrastructure, especially in remote primary care facilities.

2. Support the development of context-responsive policies that facilitate local digital innovations that place special emphasis on enhanced communication infrastructure and electronic health records.

3. Develop and implement practice-based protocols for clinical staff aimed at risk detection and early cancer diagnosis for the most common cancers.

4. Support the development of resilient, rapid transit, care pathways for patients with suspected cancer from primary care to appropriate specialist services (and back).

These recommendations point to context-specific interventions that are sensitive to the needs of local communities and health services, which could be progressively realised based on available resources.

## Strengths and Limitations

The non-random sampling, and small sample size within the geographic areas studied limits our ability to claim that these findings are generalisable to all facilities in SA, Zimbabwe or sub-Saharan Africa. The data collection process partially lent itself to bias, as participants were requested to submit facility data – the data collectors thus relied on respondent reports and not on primary data sources such as official hospital records. The possibility of bias where a single source was used is acknowledged. The strength of our approach, which enhances the validity of this study, was in the comprehensive nature of the data collected, the involvement of local respondents in identifying gaps in services, and the description of the context in which these facilities operate.

## Conclusion

This cross-sectional study aimed to assess readiness of facilities to detect and diagnose cervical, breast and colorectal cancer in two African countries. It establishes a baseline for future quality improvement interventions in these geographic areas. The key findings showed that for PC, there was variable capacity to provide early diagnosis for cervical and breast cancers, and virtually no capacity for suspected colorectal cancer. Specialised care at secondary and tertiary hospitals were better prepared for all three cancers, though the absence of mammography services in Zimbabwe is notable. Some recommendations for health system strengthening are made. Future research should explore innovations such as mobile-linked technology to enhance clinical capacity in PC, community education, and linking patients to the most appropriate care facility based on individual need.

## Supporting information

**S1 Appendix. The self-administered data collection tool used to collect all study data.**
(DOCX)

**S2 Appendix. Supplementary data tables providing full details on health facility profiles.**
(DOCX)

**S3 Appendix. Supplementary data tables on equipment needed for cancer diagnosis.**
(DOCX)

**S1 Data. Raw dataset.**
(XLSX)

**S1 Checklist. Inclusivity in global research.**
(DOCX)

## Acknowledgments

We acknowledge the contribution of other members of the AWACAN-ED team for their support in other project activities related to this study. We thank the participants for their time in interviews.

## Author contributions

**Conceptualization:** Tasleem Ras, Sarah Day, Valerie A. Sills, Suzanne E. Scott, Fiona M. Walter, Jennifer Moodley.

**Data curation:** Tasleem Ras.

**Formal analysis:** Tasleem Ras, Sarah Day.

**Funding acquisition:** Valerie A. Sills, Suzanne E. Scott, Fiona M. Walter, Jennifer Moodley.

**Investigation:** Tasleem Ras, Bothwell Guzha, Fiona M. Walter, Jennifer Moodley.

**Methodology:** Tasleem Ras, Sarah Day, Bothwell Guzha, Valerie A. Sills, Suzanne E. Scott, Fiona M. Walter, Jennifer Moodley.

**Project administration:** Fiona M. Walter, Jennifer Moodley.

**Resources:** Fiona M. Walter.

**Supervision:** Tasleem Ras, Sarah Day, Bothwell Guzha, Suzanne E. Scott, Fiona M. Walter, Jennifer Moodley.

**Validation:** Bothwell Guzha.

**Writing – original draft:** Tasleem Ras.

**Writing – review & editing:** Tasleem Ras, Sarah Day, Bothwell Guzha, Valerie A. Sills, Suzanne E. Scott, Fiona M. Walter, Jennifer Moodley.

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
