## [Decision Letter · Decision Letter 0]

17 Aug 2025

PGPH-D-25-01426

Health facility preparedness for early detection of symptomatic cancer in Southern Africa: a multi-centre cross-sectional study

Dear Dr. Ras,

Thank you for submitting your manuscript to PLOS Global Public Health. After careful consideration, we feel that it has merit but does not fully meet PLOS Global Public Health’s publication criteria as it currently stands. Therefore, we invite you to submit a revised version of the manuscript that addresses the points raised during the review process.

The manuscript has been evaluated by one reviewer, and their comments are available below.

Could you please carefully revise the manuscript to address all comments raised?

Please note that we have only been able to secure a single reviewer to assess your manuscript. We are issuing a decision on your manuscript at this point to prevent further delays in the evaluation of your manuscript. Please be aware that the editor who handles your revised manuscript might find it necessary to invite additional reviewers to assess this work once the revised manuscript is submitted. However, we will aim to proceed on the basis of this single review if possible.

We look forward to receiving your revised manuscript.

Kind regards,

Alejandro Torrado Pacheco, PhD

Staff Editor

Journal Requirements:

1. Please clarify all sources of funding (financial or material support) for your study. List the grants (with grant number) or organizations (with url) that supported your study, including funding received from your institution.

2. State the initials, alongside each funding source, of each author to receive each grant.

3. State what role the funders took in the study. If the funders had no role in your study, please state: “The funders had no role in study design, data collection and analysis, decision to publish, or preparation of the manuscript.”

4. If any authors received a salary from any of your funders, please state which authors and which funders.

3. Please send a completed 'Competing Interests' statement, including any COIs declared by your co-authors. If you have no competing interests to declare, please state "The authors have declared that no competing interests exist". Otherwise please declare all competing interests beginning with the statement "I have read the journal's policy and the authors of this manuscript have the following competing interests:"

4. In the online submission form, you indicated that The full dataset is available on request from the corresponding author at tasleem.ras@uct.ac.za.

3. Uploaded as supplementary information.

5. Please provide separate figure files in .tif or .eps format.

6. We have noticed that you have uploaded Supporting Information files, but you have not included a list of legends. Please add a full list of legends for your Supporting Information files after the references list.

Reviewers' comments:

Reviewer's Responses to Questions

**Comments to the Author**

1. Does this manuscript meet PLOS Global Public Health’s publication criteria? Is the manuscript technically sound, and do the data support the conclusions? The manuscript must describe methodologically and ethically rigorous research with conclusions that are appropriately drawn based on the data presented.? Is the manuscript technically sound, and do the data support the conclusions? The manuscript must describe methodologically and ethically rigorous research with conclusions that are appropriately drawn based on the data presented.

Reviewer #1: Yes

2. Has the statistical analysis been performed appropriately and rigorously?

Reviewer #1: No

3. Have the authors made all data underlying the findings in their manuscript fully available (please refer to the Data Availability Statement at the start of the manuscript PDF file)?

The PLOS Data policy requires authors to make all data underlying the findings described in their manuscript fully available without restriction, with rare exception. The data should be provided as part of the manuscript or its supporting information, or deposited to a public repository. For example, in addition to summary statistics, the data points behind means, medians and variance measures should be available. If there are restrictions on publicly sharing data—e.g. participant privacy or use of data from a third party—those must be specified.requires authors to make all data underlying the findings described in their manuscript fully available without restriction, with rare exception. The data should be provided as part of the manuscript or its supporting information, or deposited to a public repository. For example, in addition to summary statistics, the data points behind means, medians and variance measures should be available. If there are restrictions on publicly sharing data—e.g. participant privacy or use of data from a third party—those must be specified.

Reviewer #1: Yes

4. Is the manuscript presented in an intelligible fashion and written in standard English?

Reviewer #1: Yes

5. Review Comments to the Author

Reviewer #1: The manuscript addresses an important gap by assessing health facility preparedness for early cancer detection in Southern Africa. The study is novel for the region and uses a comprehensive health facility assessment tool aligned with WHO’s framework. The findings are relevant for policymakers and health systems planners. The manuscript is generally well-written and structured, but there are areas that would benefit from clarification, methodological strengthening, and deeper discussion.

1. Given that facility selection was purposive for primary care and exhaustive for secondary/tertiary facilities within the chosen referral pathways, how representative are the findings for the broader regions or for each country as a whole?

2. How were terms like “adequate communication infrastructure” and “community engagement” operationally defined? Was there a scoring threshold, and if so, how was it determined?

3. Given the substantial gap in colorectal cancer readiness at primary care level (only 9% providing clinical assessment), how do the authors recommend prioritizing this against breast and cervical cancer services which already have some infrastructure?

4. Minor style issue: Define ST (secondary/tertiary) in the abstract for readers unfamiliar with the term.

5. The recommendations include mHealth innovations, but implementation feasibility in low-connectivity areas is challenging. Did the authors assess any existing successful pilots in similar contexts that could be referenced?

6. PLOS authors have the option to publish the peer review history of their article (what does this mean?). If published, this will include your full peer review and any attached files.). If published, this will include your full peer review and any attached files.

**Do you want your identity to be public for this peer review?** For information about this choice, including consent withdrawal, please see our Privacy Policy..

Reviewer #1: No

---

## [Decision Letter · Decision Letter 1]

2 Feb 2026

PGPH-D-25-01426R1

Health facility preparedness for early detection of symptomatic cancer in Southern Africa: a multi-centre cross-sectional study

Dear Dr. Tasleem Ras

Thank you for submitting your manuscript to PLOS Global Public Health. After careful consideration, we feel that it has merit but does not fully meet PLOS Global Public Health’s publication criteria as it currently stands. Therefore, we invite you to submit a revised version of the manuscript that addresses the points raised during the review process.

We look forward to receiving your revised manuscript.

Kind regards,

Martin Mickelsson, Ph.D.

Academic Editor

Journal Requirements:

Additional Editor Comments (if provided):

Reviewers' comments:

Reviewer's Responses to Questions

**Comments to the Author**

1. If the authors have adequately addressed your comments raised in a previous round of review and you feel that this manuscript is now acceptable for publication, you may indicate that here to bypass the “Comments to the Author” section, enter your conflict of interest statement in the “Confidential to Editor” section, and submit your "Accept" recommendation.

Reviewer #2: All comments have been addressed

Reviewer #3: (No Response)

2. Does this manuscript meet PLOS Global Public Health’s publication criteria? Is the manuscript technically sound, and do the data support the conclusions? The manuscript must describe methodologically and ethically rigorous research with conclusions that are appropriately drawn based on the data presented.? Is the manuscript technically sound, and do the data support the conclusions? The manuscript must describe methodologically and ethically rigorous research with conclusions that are appropriately drawn based on the data presented.

Reviewer #2: Yes

Reviewer #3: Yes

3. Has the statistical analysis been performed appropriately and rigorously?

Reviewer #2: Yes

Reviewer #3: (No Response)

4. Have the authors made all data underlying the findings in their manuscript fully available (please refer to the Data Availability Statement at the start of the manuscript PDF file)?

The PLOS Data policy requires authors to make all data underlying the findings described in their manuscript fully available without restriction, with rare exception. The data should be provided as part of the manuscript or its supporting information, or deposited to a public repository. For example, in addition to summary statistics, the data points behind means, medians and variance measures should be available. If there are restrictions on publicly sharing data—e.g. participant privacy or use of data from a third party—those must be specified.requires authors to make all data underlying the findings described in their manuscript fully available without restriction, with rare exception. The data should be provided as part of the manuscript or its supporting information, or deposited to a public repository. For example, in addition to summary statistics, the data points behind means, medians and variance measures should be available. If there are restrictions on publicly sharing data—e.g. participant privacy or use of data from a third party—those must be specified.

Reviewer #2: Yes

Reviewer #3: Yes

5. Is the manuscript presented in an intelligible fashion and written in standard English?

Reviewer #2: Yes

Reviewer #3: Yes

6. Review Comments to the Author

Reviewer #2: Dear Authors,

I appreciate you taking up this topic and your study is an important one.

I have reviewed the paper and your letter to prior reviewers. You have addressed all concerns adequately. Thank you

Reviewer #3: The manuscript presents a timely and well structured analysis of existing gap by the evaluating health facilities preparedness for early cancer detection is selected countries of Southern Africa. The study is clearly written, methodologically sound, and contributes valuable insights to the field. Overall, the paper is interesting and acceptable for publication, pending minor revisions.

- In the abstract, when a term appears only once, there is no need—and no requirement—to introduce its abbreviation. The same principle applies throughout the manuscript (e.g., line 41 and line 385).

- Minor technical issue: the abstract lacks consistency in how numerical information is presented. Please use a uniform format, spelling out numbers with numerals in parentheses for all relevant data.

- The nurse‑to‑1000‑population ratios for primary care are inconsistent between the abstract and the Results section. These figures should be aligned.

- How are PC facilities distributed across the selected regions? It would be helpful to indicate the distances between these facilities, which would improve readers’ understanding of geographical accessibility.

- It would strengthen the manuscript to include findings related to medical record systems. This topic should be incorporated into the Results and Discussion section as well, particularly in the context of future recommendations and policy implications. It would be valuable to comment on the current state policy in this area as well.

- Line 395: The reference to “Sustainable Development Goal requires clarification. Its relevance is not immediately clear, especially not reflecting in Introduction section, what are targets per country for this specific issue.

- In the Discussion, the content in lines 378–380 would be more effective if moved to the end of the section and reframed as policy implications and recommendations. Additional recommendations should also be consolidated in this section, followed by a concluding paragraph to ensure a coherent and logical flow.

- The title of section in line 445 should be Strength and Limitations

7. PLOS authors have the option to publish the peer review history of their article (what does this mean?). If published, this will include your full peer review and any attached files.). If published, this will include your full peer review and any attached files.

**Do you want your identity to be public for this peer review?** For information about this choice, including consent withdrawal, please see our Privacy Policy..

Reviewer #2: **Yes:** Shailendra PrasadShailendra Prasad

Reviewer #3: No

 Figure Resubmissions:

---

## [Editor Report · Decision Letter 2]

19 Mar 2026

Health facility preparedness for early detection of symptomatic cancer in Southern Africa: a multi-centre cross-sectional study

PGPH-D-25-01426R2

Dear Tasleem Ras,

We are pleased to inform you that your manuscript 'Health facility preparedness for early detection of symptomatic cancer in Southern Africa: a multi-centre cross-sectional study' has been provisionally accepted for publication in PLOS Global Public Health.

Best regards,

Martin Mickelsson, Ph.D.

Academic Editor
